# Development of an Analytical Model for Optimization of Direct Laser Interference Patterning

**DOI:** 10.3390/ma13010200

**Published:** 2020-01-03

**Authors:** Bogdan Voisiat, Alfredo I. Aguilar-Morales, Tim Kunze, Andrés Fabián Lasagni

**Affiliations:** 1Institut für Fertigungstechnik, Technische Universität Dresden, George-Bähr-Str. 3c, 01069 Dresden, Germany; andres_fabian.lasagni@tu-dresden.de; 2Fraunhofer-Institut für Werkstoff- und Strahltechnik IWS, Winterbergstr. 28, 01277 Dresden, Germany; alfredo.morales@iws.fraunhofer.de (A.I.A.-M.); tim.kunze@iws.fraunhofer.de (T.K.)

**Keywords:** direct laser interference patterning, microstructures, process optimization, analytical model, laser ablation, stainless steel, line-like structure, microstructure

## Abstract

Direct laser interference patterning (DLIP) has proven to be a fast and, at the same time, high-resolution process for the fabrication of large-area surface structures. In order to provide structures with adequate quality and defined morphology at the fastest possible fabrication speed, the processing parameters have to be carefully selected. In this work, an analytical model was developed and verified by experimental data, which allows calculating the morphological properties of periodic structures as a function of most relevant laser-processing parameters. The developed model permits to improve the process throughput by optimizing the laser spot diameter, as well as pulse energy, and repetition rate. The model was developed for the structures formed by a single scan of the beam in one direction. To validate the model, microstructures with a 5.5 µm spatial period were fabricated on stainless steel by means of picosecond DLIP (10 ps), using a laser source operating at a 1064 nm wavelength. The results showed a difference of only 10% compared to the experimental results.

## 1. Introduction

Direct laser interference patterning (DLIP) is a well-established technology for large-area surface treatment. The method provides fast patterning of periodic structures with high resolution (even below 1 µm) [1,2]. Such microstructures have been intensively investigated, because of their potential utilization for surface functionalization in different application fields [3,4,5]. For example, surface functions including tribological, superhydrophobicity, and bacteria-repellency have been already demonstrated by means of DLIP in previous works [6,7,8]. It was also showed that this technology can be used for decorative applications by forming structural colors on the material surface, including in steels and polymers [9,10,11].

For the treatment of large surface areas, the fabrication speed (throughput) becomes a very important factor, especially in mass production. In the case of laser-based structuring processes, the power and the repetition rate of the laser source mainly determine the maximal possible processing speed [12]. For example, in the case of the DLIP technology, optical configurations permitting to shape the laser beam allowed record processing speeds of 0.36 m²/min and 0.9 m²/min on metal and polymers, respectively [13,14]. In the last case, the average laser power used was 155 W, using nanosecond pulses with a pulse duration of 8 ns. 

In addition, it is also known that the structuring speed highly depends on the structure depth of the features, since deeper structures require more pulses per spot, which results in a reduction of the throughput since a larger amount of energy doses are needed [15,16]. The laser fluence is also a very important parameter that governs the final fabricated structure morphology in DLIP. For example, D’Alessandria et al. showed for metals the existence of a critical laser fluence, which determines the characteristic structure formation mechanisms, namely Marangoni convection or recoil pressure [17]. It has also been shown that the laser fluence together with the distance between laser pulses (pulse to pulse overlaps) not only affects the structure’s depth but also its quality. For example, Aguilar-Morales et al. studied the quality of the DLIP-fabricated structures in terms of their homogeneity across the whole patterned area, showing that careful optimization of processing parameters is needed to achieve a uniformly distributed structure on the whole patterned area [18]. Overall, there are many processing parameters, such as laser fluence, frequency, scan/translation speed, distance between pulses (overlap), and hatch-distance, which in different ways influence both the final geometry and depth, as well as the fabrication speed of the structures produced in the DLIP method.

This work introduces an analytical model to optimize the DLIP process parameters for achieving the maximal possible fabrication speed for a target structure depth. The presented analytical model is compared to experimental results carried out on stainless steel by two-beam DLIP with picosecond (10 ps) laser radiation at 1064 nm of laser wavelength. Only a one-dimensional case, meaning that the laser beam scanned in only one direction, was investigated. The fabricated line-like periodic structures were characterized by scanning electron and confocal microscopy. The obtained experimental results were used to validate the model and predict the optimal parameters for the fastest possible DLIP process, depending on the available average laser power.

## 2. Materials and Methods 

### 2.1. Materials

The DLIP experiments were performed on 0.7 mm thick metal plates made of X5CrNi18-10 stainless austenitic chromium–nickel steel (also called 1.4301). The surface of the plates was electro-polished providing a surface roughness Ra below 60 nm. Prior to the laser process, the substrates were cleaned from contaminations using ethanol. 

### 2.2. Experimental Setup 

The structuring of the steel samples was performed using a two-beam DLIP configuration. A picosecond (10 ps) laser source (Innoslab Nd:YVO4, EdgeWave GmbH, Würselen, Germany) operating at a wavelength of 1064 µm and a repetition rate up to 30 kHz was used. The laser emits a Gaussian beam with the beam quality of M² = 1.1 and the high pulse energy stability of 1.1 RMS. This is the most common type of laser source, which is also used in the industry for laser material microfabrication. The laser beam was directed with mirrors to the DLIP optical head (Fraunhofer IWS, Dresden, Germany) where the beam was split into two beams using a diffractive optical element (DOE). The split beams were then parallelized by a prism and finally overlapped using a converging lens with a focal distance of 40 mm. By varying the distance between the prism and the DOE, the spatial period of the created interference intensity profile can be controlled in the range of 1.5 µm to 5.5 µm. The detailed description of the positioning system and optical setup used in the DLIP-module can be found in [15,19]. The DLIP module was fixed on a vertical motorized stage (ATS100, Aerotech, Inc., Pittsburgh, PA, USA), which was used to set the accurate working distance between the head and the sample. The fabricated structure was extended in the perpendicular direction to the structure period by shifting the samples in the *y*-direction with a motorized stage (ABL1500, Aerotech, Inc., Pittsburgh, PA, USA) as shown in Figure 1. The structure was not extended along the *x*-direction in order to eliminate the aspect of potential positioning inaccuracy that can affect the final structure height. This happens when the stitching of the structures in the *x*-direction is not accurate enough (is not equal to the integer number of periods). In this case, the forthcoming pulse (in the *x*-direction) destroys the already-formed structure, reducing the final structure height. Therefore, in order to eliminate such a possibility, the structure was extended by scanning the beam only in the *y*-direction.

The laser beam power was measured using a power meter (Nova ΙΙ, Ophir Optronics Solutions Ltd., Jerusalem, Israel) equipped with a thermal power sensor (30A-BB-18, Ophir). In addition, the laser spot size (at the intensity level of 1/e²) was measured on the sample surface using a beam-profiling camera (BEAMAGE-4M-IR, Gentec Electro-optics, Inc., Quebec city, Canada). Figure 2 shows the results of one of the measurements performed using the beam-profiling camera. The beam size was calculated as an average of the horizontal and vertical beam sizes. 

In the area where the two coherent laser beams (with a beam radius ω) are superimposed at an angle θ, a periodical line-like intensity distribution is created on the sample surface with a determined spatial period Λ (see Figure 2). The resulting interference profile can be utilized for direct ablation of the steel sample surface to form a so-called line-like periodic structure. By shifting the laser spots in the *y*-direction (see Figure 2) with a defined speed v it is possible to treat larger areas. Because the laser pulses are emitted at a certain repetition rate f, the forthcoming laser spots become displaced by a distance d, which is calculated accordingly:(1)$$d=\frac{v}{f}.$$

To describe how much the spots overlap (OVP), it is possible to calculate this value as a function of the beam radius ω as well as the distance between the pulses d, obtaining:(2)$$OVP=1-\frac{d}{2\omega }.$$

Thus, the spots’ displacement distance d can also be expressed in terms of the OVP:(3)d=2ω(1−OVP).

The beam radius corresponds to the radius of the spot formed on the samples’ plane. In principle, the spot on the sample is not circular but rather elliptical. However, if the interception angle between the beams (θ) is small, the ellipticity of the spot can be neglected. In this work θ = 11.2°, which results only in 2% ellipticity (see Figure 1). Therefore, the spot radius is considered the same in both *x* and *y* directions. 

### 2.3. Surface Characterization

A scanning electron microscope (SEM) (ZEISS Supra 40VP, Jena, Germany) was used to visualize the surface morphology of the laser-treated samples. The topography of the microstructures was also analyzed using a confocal microscope (Sensofar S neox, Terrassa, Spain) equipped with a 150× magnification objective with a resolution of 170 nm and 2 nm in the lateral and vertical directions, respectively. 

## 3. Results and Discussion

### 3.1. Development of an Analytical Ablation Model for DLIP

An analytical laser-ablation model, describing the DLIP-fabricated structure depth dependence on laser process parameters was developed in this work. The graphical representation of the model is depicted in Figure 3. The interfering laser spot is scanned along the interference lines on a surface (along the *y*-direction). Each pulse ablates locally the surface of the material at the intensity maximal positions forming a periodic structure across the area of radius $r_{th}$. Assuming that the overlapped laser beam has a perfect Gaussian spatial beam profile, the structured area diameter can be calculated using Equation (4) [20]:(4)$$r_{th}=\omega \sqrt{\frac{1}{2}\ln\left(\frac{F_{0}}{F_{th}}\right)},$$
where $F_{0}$ is the peak fluence of the Gaussian beam and $F_{th}$ is the ablation threshold fluence. The ablated structure depth can be described by simple logarithmic dependence, which has been already used to describe photochemical or photothermal ablation processes (which is suitable for many materials) as described by Equation (5) [21]:(5)$$\Delta h=\delta \ln\left(\frac{F}{F_{th}}\right),$$
where δ is the energy penetration depth. 

The ablated structure depth will be calculated in the center of the Gaussian spot (point of interest, represented by the green dot in Figure 3a). During the structure formation process, the forthcoming spots overlap. Thus, the areas in the overlap regions are exposed by multiple pulses, resulting in deeper structures. Therefore, the final structure depth can be calculated by adding all depths ablated by each pulse within the radius $r_{th}$ from the point of interest. However, it has to be noted that each pulse in the scan row ablates the point of interest with different local fluences F, due to the Gaussian spatial beam profile (see Figure 3b). Thus, it is important to calculate the fluence as a function of the pulse number:(6)$$F\left(n\right)=F_{0}\exp\left(-2\frac{r{\left(n\right)}^{2}}{\omega^{2}}\right),$$
where $F_{0}$ is the laser peak fluence and r(n) is the distance from the point of interest to the center of the pulse, which is expressed in terms of pulse number n and the total number of pulses N within the structured area diameter $r_{th}$ (see Figure 3c) using Equations (7) and (8):(7)$$r\left(n\right)=d\left(-\frac{N-1}{2}+n-1\right)=\frac{d}{2}\left(2n-N-1\right)=\omega \left(1-OVP\right)\left(2n-N-1\right),$$
(8)$$N=2\cdot floor\left(\frac{r_{th}}{d}\right)+1=2\cdot floor\left(\frac{\sqrt{\frac{1}{2}\ln\left(\frac{F_{0}}{F_{th}}\right)}}{2\left(1-OVP\right).}\right)+1.$$

Here floor is a standard mathematical floor function, which takes as input a real number and gives as output the greatest integer less than, or equal to, the input number. This is done in order to have an integer number of interest points.

By substituting the Equations (6)–(8) into Equation (5), the ablated depth for a defined number of pulses is obtained. By summing all the depths ablated by each pulse at the point of interest, the total ablation depth can be calculated by means of Equation (9):(9)$$h=\delta \overset{N}{\underset{n=1}{\sum }}\ln\left(\frac{F\left(n\right)}{F_{th}}\right).$$

Finally, by substituting the spot displacement with OVP and simplifying Equation (9), the following expression for the final ablated depth is obtained:(10)$$h=\delta \cdot N\left(\ln\left(\frac{F_{0}}{F_{th}}\right)-\frac{2}{3}{\left(1-OVP\right)}^{2}\left(N^{2}-1\right)\right).$$

The laser peak fluence $F_{0}$ for the two-beam interference can be also expressed by the average laser power P, laser repetition rate f and radius of the interference spot ω as follows:(11)$$F_{0}=\frac{2\cdot 2\cdot P}{f\cdot \pi \cdot \omega^{2}}.$$

As can be seen in Equation (11), an additional multiplication factor of 2 was added, due to the two-beam interference pattern [22]. It must be highlighted that the ablation threshold used in these calculations does not depend on the pulse number, which means that the incubation effect is neglected and $F_{th}$ represents the effective ablation threshold for the set of pulses applied in the experiment [23].

### 3.2. Validation of the Simulation Model

In order to verify the developed ablation model from Section 3.1, periodic line-like structures with a fixed spatial period of Λ = 5.5 µm were fabricated on a stainless steel sample. The maximal period was selected, because of the limited lateral resolution of the confocal microscopy. The measurement of the height of the structures with bigger feature sizes is more accurate, and hence a more precise verification of the proposed model can be performed. For the structuring process, the repetition rate was set to 1 kHz and the maximum laser power was utilized corresponding to a pulse energy of 0.44 mJ at the sample surface. Such a low repetition rate was chosen in order to eliminate the heat accumulation that usually appears at high repetition rates and also affects the experimental results [24]. The laser fluence was changed by varying the spot diameter between 160 µm and 285 µm that resulted in peak fluences between 8.8 and 2.8 J/cm², respectively. The structures were fabricated by varying not only the laser spot diameters but also the pulse overlap, namely OVP = 90, 91, 93, 95, 97 and 99%.

SEM and confocal microscopy images of the structures fabricated on stainless steel with 95, 97 and 99% overlap, using a spot diameter of 285 µm (0.56 J/cm²), are shown in Figure 4. The micrographs were taken at the center of the scanned path (see Figure 2). As can be seen, the structures contain periodically distributed trenches, the depth of which increases with higher overlaps. In all cases, the interference microstructure is covered with the finer periodic structures, which are called laser-induced periodic surface structures (LIPSS). The formation of LIPSS on top of the DLIP structure was already reported in [15,25]. Due to the characteristic of the structures, they could be classified as low spatial frequency LIPSS with a spatial period of $\Lambda_{LIPSS}$ = 850 nm, which is similar to the used laser wavelength.

After laser patterning experiments, the mean depths, h, of the fabricated structures were measured using confocal microscopy. The depth h was measured in the center of the structured area (see Figure 2) and averaged across the area of 25 × 25 µm², which is the field of view of the utilized 150× objective. The three-dimensional scatter plot of the measured depths of all fabricated structures is depicted in Figure 5a as black dots with drop lines.

The measured depth values were then compared with the developed model based on Equation (10). The values for energy penetration depth (δ =8.9 nm) and the ablation threshold ($F_{th}$ = 0.055 J/cm²) that were used for the calculations were taken from [26]. The calculated results are plotted as a green surface in Figure 5a, together with the experimental data points. The relative residual error of the model is plotted in Figure 5b. The difference between the modeled and the measured depths ranges between −12% and 20%, resulting in an average error of 7.4%, which shows that the model is in good agreement with the measured values. 

This model can be used to determine the parameter sets (overlap OVP and diameter of the interference spot 2w) corresponding to a defined structure depth. For this reason, Equation (10) was numerically solved to express the overlap (OVP) as a function of the interference spot size (2w) for a fixed structure depth. The resulting curves showed in Figure 6a represent the isocurves of the surface plotted in Figure 5a. Each point in the isocurve corresponds to the combination of OVP and 2*w*, resulting in the same structure depth (e.g. 0.5 µm; 1.0 µm; 1.5 µm, etc.). The results show that there are many different combinations of parameters (spot radius or laser fluence, and spot overlap), which can be used to produce patterns with the same structure depth. However, the throughput for each combination of parameters might be different. Therefore, it is important to determine which set of parameters is optimal for optimizing the fabrication speed (or throughput).

For this reason, the fabrication speed (area per unit of time) was calculated using the following equation:(12)$$\frac{\partial A}{\partial t}=2r_{th}v=2r_{th}df=\frac{16P}{F_{0}\pi }\left(1-\frac{OVP}{100\%}\right)\sqrt{\frac{1}{2}\ln\left(\frac{F_{0}}{F_{th}}\right)}.$$

The resultant plot showing this equation is depicted in Figure 6b. All curves in the diagram show that the maximal throughput is reached when the spot diameter is about 746 µm. This diameter is more than two times bigger than the spot sizes used in the DLIP experiments. It means that the process parameters used in the experiments are not optimal in terms of structuring speed. For example, the patterns with 500 nm structure depth were fabricated with a spot size of 285 µm at 93% overlap, resulting in a fabrication speed of 4.7 cm²/min at 1 kHz of repetition rate (black dot in Figure 6a,b). However, according to the model, the increase of the interference spot size and overlap to their optimal values of 746 µm and 97.6%, respectively, would result in an increase of structuring speed up to 7.9 cm²/min, which is approximately 1.7 times faster (see the blue dot in Figure 6a,b).

In Figure 7, the modeled maximal structuring speed (blue line) is plotted together with the experimentally achieved throughputs (empty circles) with respect to the structure depth between 0 and 5 µm. As can be seen, the modeled throughput is always higher than the throughputs reached experimentally, because the spot size used in experiments was smaller than the optimal spot size estimated using the proposed model. To verify if the predicted throughputs can be reached experimentally, the laser spot size should be increased to 746 µm in diameter. However, the optical setup used in experiments does not provide such ability, since the maximal settable spot size in the system is only 285 µm. Therefore, the calculated optimal conditions for the fastest structuring process were not verified experimentally.

## 4. Conclusions

In conclusion, an analytical model that can be used to calculate the structure depth of line-like periodic structures fabricated using DLIP was developed. Using a two-beam setup, the structure depth was varied from 0.5 µm to 4.9 µm, by controlling the laser fluence and the pulse overlap. The comparison of the structure depth between the fabricated microstructures and the calculated values showed a difference below 10%. The model also permitted the optimization of the process parameters to reach a certain structure depth at the maximal possible throughput. However, the suggested model was tested only at a relatively narrow parameter range. Therefore, further experiments with a wider range of parameters (in particular at higher laser power levels) have to be performed in the future to validate and improve the proposed model.

## Figures and Tables

**Figure 1 materials-13-00200-f001:**
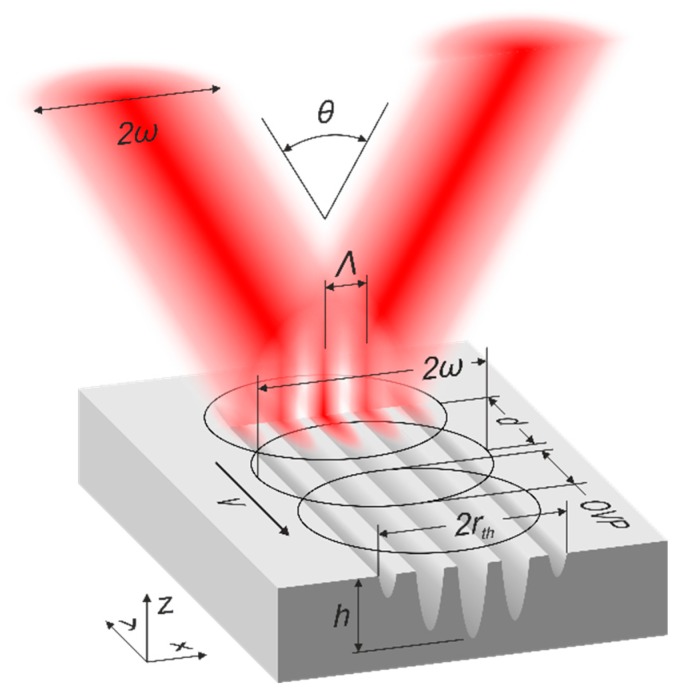
Graphical representation of the used two-beam direct laser interference patterning (DLIP) process, showing the most relevant parameters describing both the pattern and the process: ω—beam radius; θ—beams intersection angle; Λ—laser intensity profile/ ablated structure period; d—distance between the laser spots; OVP—spot overlap; v—samples’ translation speed; rth—radius of ablated area; h—ablated structure depth.

**Figure 2 materials-13-00200-f002:**
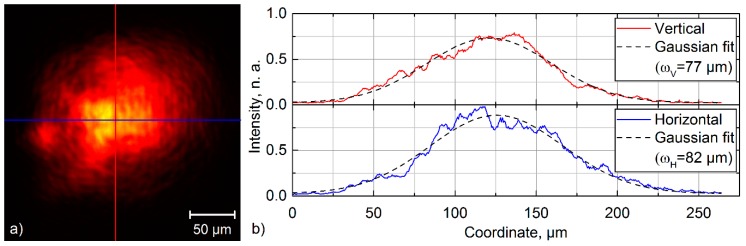
(**a**) Intensity profile of one of two interfering beams measured with a beam-profiling camera on the samples’ surface; (**b**) vertical and horizontal cross-sections of the laser intensity profile—$\omega_{H}$, $\omega_{V}$—correspond to the horizontal and vertical spot radius derived from the Gaussian fit, respectively.

**Figure 3 materials-13-00200-f003:**
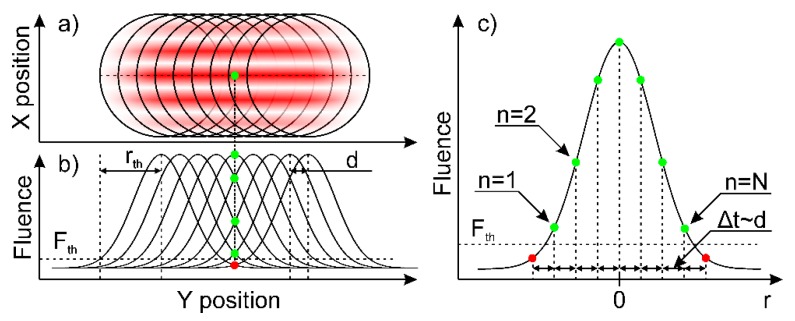
Representation of the analytical model used to calculate the depth of the DLIP-ablated line-like structure. (**a**) Graphical representation of the positions of the interfering laser spots during the scanning process. The green dot represents the position where the depth is calculated using the analytical model (point of interest). (**b**) Schematic representation of the fluence distribution of each laser spot along the center of the scanning path. Green and red dots represent the fluence, at which the point of interest was ablated with each passing pulse during laser scanning. (**c**) Laser spot center intensity profile along the scanning direction. The dots represent the laser fluence at which the point of interest was ablated by each forthcoming pulse. The green and red color of the dots corresponds to fluences higher and lower than the ablation threshold, respectively.

**Figure 4 materials-13-00200-f004:**
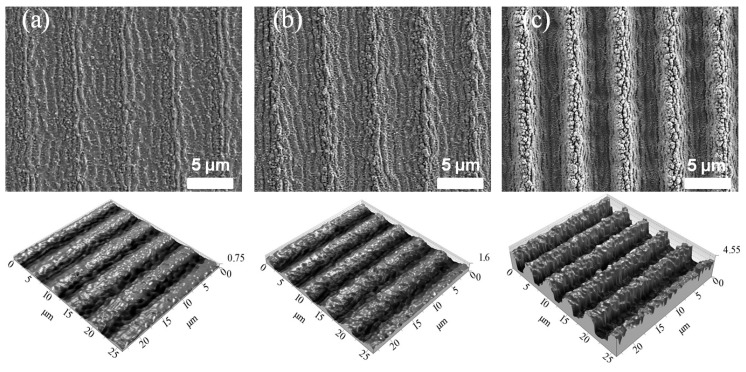
SEM (top row) and confocal (bottom row) microscopy micrographs of periodic structures on stainless steel fabricated with (**a**) 95%, (**b**) 97%, and (**c**) 99% overlap, and a spot diameter of 285 µm (0.56 J/cm²). The micrographs were taken in the center of the scanned path described in Figure 1.

**Figure 5 materials-13-00200-f005:**
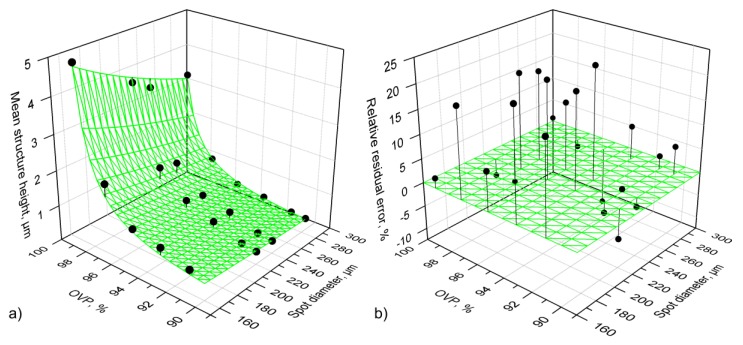
(**a**) Structure depth (measured in the center of the scanned path) dependence on the process parameters, namely spot overlap (OVP) and diameter. The black dots with drop lines represent the measured values (measured by confocal microscopy). The green surface corresponds to the structure depth calculated using the analytical model described by Equation (10). Energy penetration depth δ = 8.9 nm and the ablation threshold $F_{th}$ = 0.055 J/cm² were used in the calculation; (**b**) Relative residual error of the model plotted in Figure 5a.

**Figure 6 materials-13-00200-f006:**
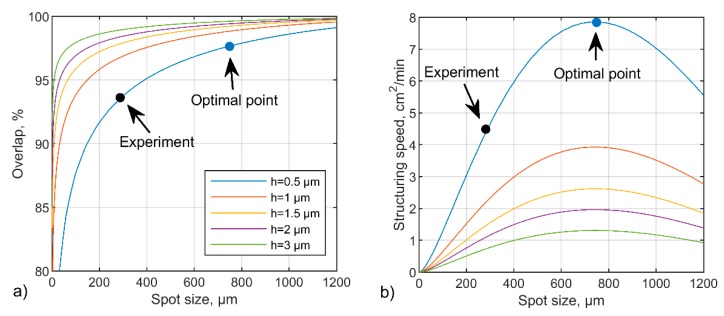
(**a**) Calculated isocurves representing the set of processing parameters, namely spot overlap (OVP) and spot size (2w) that result in definite structure depths; (**b**) calculated structuring speed for each isocurve in (**a**).

**Figure 7 materials-13-00200-f007:**
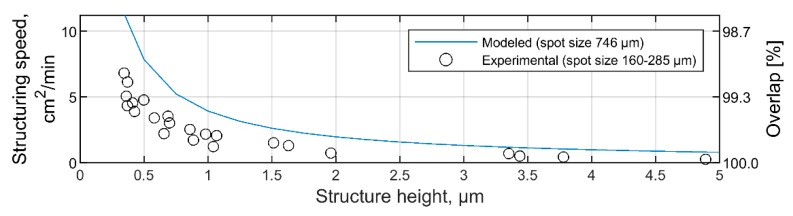
Maximal calculated (**blue line**) and experimental (**empty circles**) structuring speeds for a certain structure depth. All plots were calculated for a laser pulse energy of 0.44 mJ.

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
