# Peer review of "Development of an Analytical Model for Optimization of Direct Laser Interference Patterning"

_materials, 2020, doi:10.3390/ma13010200_

Round 1

Reviewer 1 Report

The manuscript presents an analytical model of optimization of the DLIP process parameters such as the laser spot diameter, pulse energy, and repetition rate. The presented analytical model was compared to experimental results carried out on stainless steel by one-dimension scanning case of two-beam DLIP with (10 ps) laser radiation at 1064 nm of laser wavelength. The experimental results showed disagreement with the modal of about 10%. Generally the work is interesting to be reported, however, the manuscript seems to be written in haste as there are many spelling, linguistic and even organizational errors. Here are some comments to the authors for improvements:

The abstract must be updated in order to mention that the modal was compared experimentally with two-beam DLIP under the case of only one-dimension scanning, as stated in the introduction. The manuscript should be rewritten carefully and to standardize the title/number of references for figures. E.g. (“Fig”.1) (“Figure” 1). There is no “Figure 5c” in the manuscript as mentioned on page 7,line 225, but figure 6. Meanwhile figure 5(a,b) mentioned as figure 6 (a,b). Section 2, Materials and Methods, 2.1. Materials, This part must be deleted. It seems it left from the journal paper template. This is a simple example to confirm that the manuscript wasn't checked carefully prior to the submission. How about the power stability of the laser source? How do the power fluctuations of the laser affect the achieved results?. This must be discussed in the manuscript. Validation of the simulation model was done on microstructures with a 5.5 μm spatial period, where the laser repetition rate was only set to 1 kHz. Although the laser source (Innoslab Nd:YVO4) operating at a wavelength of 1064 μm and a repetition rate of up to 30 kHz, there were no investigations on ablation under different repetition rates. The average error between the calculated depth and the experimental data reported as 10 %, thus, the repeatability of the process with respect to the achieved results should be addressed in the manuscript. Check the spelling of the vertical axis title of figure 5b.

Reviewer 2 Report

File attached.

Reviewer 3 Report

In this paper entitled ”Development of an analytical model for optimization of Direct Laser Interference Patterning” the authors study the use of DLIP to texture large-area surface structures. The manuscript is interesting and details are well explained. Only a few points have to be addressed therefore a minor revision should be done.

Comments:

Page 1: Why the model has been validated with microstructures with 5.5 µm period, 10 ps and 1064 nm wavelength and 30 kHz?. Page 2, line 59: Why the laser beam is scan in only one direction?. Page 2, lines 65 to 76. This text does not seem related to the paper. Please, check. Page 7, line 198: the author comment that the “average error between the calculated depth and the experimental data was only of about 10%”, however, in the Conclusions it is stated that it is below 10 %. Is this the average difference between the 23 experimental points and the calculated data of Fig. 4?. What is the maximum and minimum difference and for what conditions (structure height, OVP and spot diameter”?.

Round 2

Reviewer 2 Report

The authors have comprehensively responded to all points raised and made the paper significantly clearer and more correct. The previous paper was unclear from the start about axis/axes of motion, which could cause a reader to misinterpret Figure 3 and hence eqs (7),(8), but the authors have made the necessary changes.

Figure 3 could maybe be made clearer by showing how points in (b), some of which are superimposed, correspond to points in (c), but this is optional.